# Towards High Surface Area α-Al_2_O_3_–Mn-Assisted Low Temperature Transformation

**DOI:** 10.3390/ma16083047

**Published:** 2023-04-12

**Authors:** Tim Jähnichen, Simon Carstens, Maximilian Franz, Otto Laufer, Marianne Wenzel, Jörg Matysik, Dirk Enke

**Affiliations:** 1Institute of Chemical Technology, Leipzig University, Linnéstr. 3, D-04103 Leipzig, Germany; 2Institute of Analytical Chemistry, Leipzig University, Linnéstr. 3, D-04103 Leipzig, Germany

**Keywords:** manganese-assisted synthesis, low-temperature α-Al_2_O_3_, high specific surface area of 56 m^2^ g^−1^

## Abstract

When impregnated with manganiferous precursors, γ-Al_2_O_3_ may be converted into α-Al_2_O_3_ under relatively mild and energy-saving conditions. In this work, a manganese assisted conversion to corundum at temperatures as low as 800 °C is investigated. To observe the alumina phase transition, XRD and solid-state ^27^Al-MAS-NMR are applied. By post-synthetical treatment in concentrated HCl, residual manganese is removed up to 3 wt.-%. Thereby, α-Al_2_O_3_ with a high specific surface area of 56 m^2^ g^−1^ is obtained after complete conversion. Just as for transition alumina, thermal stability is an important issue for corundum. Long-term stability tests were performed at 750 °C for 7 days. Although highly porous corundum was synthesized, the porosity decreased with time at common process temperatures.

## 1. Introduction

Aluminum oxide is widely employed for catalytic applications, most commonly in its γ-form. γ-Al_2_O_3_ is readily available, easily moldable, e.g., for wash coating, and its porosity can be precisely controlled. In addition, the material has large specific surface areas in the range of 150 to 300 m^2^ g^−1^ and attainable pore sizes in the mesopore and macropore range [1]. For certain applications, catalyst supports need to endure high temperatures above 1000 °C [2]. This includes, among others, petroleum refining processes like steam reforming [3,4], combined heat and power units (CHP) [5], and wood-burning stoves [6]. While γ-alumina is a suitable material for most catalytic applications, it is subjected to an irreversible phase transition to thermodynamically stable α-modification at temperatures above 1000 °C [7,8].

This irreversible transition from γ- to α-Al_2_O_3_ entails several problems. As the underlying crystal structure fundamentally changes from cubic to hexagonal, a complete rearrangement of the entire alumina microstructure occurs. An unwanted but inevitable side effect thereof is the loss of porosity, and hence of internal surface area [9]. This could possibly be circumvented by substituting γ-Al_2_O_3_ with the thermodynamically stable α-modification. However, the application of α-Al_2_O_3_ is difficult because the material usually has a very low specific surface area, which diminishes catalyst loadings and dispersion, greatly impairing its application as catalyst support. Many attempts have been undertaken to introduce porosity into intrinsically non-porous corundum [10]. One approach to preserving γ-Al_2_O_3_ porosity consists of lowering the α-transition temperature, which is also less costly and saves energy. This may be achieved, for instance, by impregnating γ-Al_2_O_3_ catalyst supports with a manganiferous precursor, as reported by Tsyrulnikov et al. [11]. Upon subsequent drying, Mn(II)-ions are dispersed on the surface of the transition alumina. Calcination at 500–800 °C leads to solid solutions of the now oxidized Mn(III)-ions in the Al_2_O_3_ structure. These Mn-doped corundum-type domains act as seeding crystals, resulting in the complete conversion of the transition alumina into α-Al_2_O_3_ at 900 °C. We recently applied this procedure to sol-gel samples [12,13]. While pure alumina samples require a temperature of 1200 °C for complete α-transition, this was attained at temperatures as low as 900 °C after Mn-impregnation. Successive extraction of manganiferous species by dissolution in aqua regia yields α-Al_2_O_3_ with an increased A_BET_ of 23 m^2^ g^−1^. This is a remarkably high surface area for an α-Al_2_O_3_ without any traceable transition alumina modifications.

Most often, the phase composition, or more precisely the complete transition to corundum, is monitored by XRD. In this work, in addition to XRD, solid-state ^27^Al-MAS-NMR is used as a complementary method to observe phase transitions. The ^27^Al isotope is a highly sensitive NMR nucleus yielding broad lines over a wide chemical shift range [14]. As a spin 5/2 nucleus, it is quadrupolar. Hence, the line width is dependent on the structure, where highly symmetric environments lead to narrower lines. An issue regarding the ^27^Al-NMR spectra is whether these spectra show intensities properly, representing the proportional amounts of different aluminum species in the samples. This is particularly true for ^27^Al-NMR signals associated with aluminum sites with very low symmetry, such as θ- and γ-alumina [15]. In general, ^27^Al-MAS-NMR is used to detect the presence of aluminum and measure its relaxation time to elucidate binding relations. However, there are also studies regarding different aluminum structures and their conversions [15,16,17,18]. In 1997, Fitzgerald et al. [15] used ^27^Al-MAS-NMR for dehydration studies of high surface area alumina. They published spectra of γ- and α-Al_2_O_3_ alongside various aluminum oxide materials and were able to determine a difference between 6-fold (9 ppm), 5-fold (37 ppm), and 4-fold (69 ppm) coordinated Al sites. In a later work, O’Dell et al. [18] investigated the conversion from boehmite to corundum from RT to 1200 °C. Thereby, they were able to show that different γ-, δ- and θ-alumina phases could be detected by NMR with increasing synthesis temperature, while the material was X-ray amorphous. The group assumed that the previous alumina phases were X-ray amorphous, due to small nanocrystal sizes between 10 to 20 nm, before grain growth occurs during the transition to corundum [19,20]. Hence, they recognized ^27^Al-MAS-NMR as an important method in studying phase transitions in alumina.

In this publication, we investigated the Mn-assisted α-transition of sol-gel alumina materials, using ^27^Al-MAS-NMR and XRD as complementary methods. Thereby, we were able to show that the temperature threshold for a complete formation of α-Al_2_O_3_ is as low as 800 °C and results in a specific surface area A_BET_ of 56 m^2^ g^−1^, which is an unusually high value for corundum. Moreover, we investigated the long-term thermal stability of the thus obtained α-Al_2_O_3_. Here, we observed a degradation in specific surface area with prolonged exposure at 750 °C, similar to observations for diaspore-derived corundum [21,22,23].

## 2. Materials and Methods

### 2.1. Preparation of Alumina Starting Material

To ensure all samples originated from the exact same batch, a large batch of γ-Al_2_O_3_ was prepared, similar to the previously reported method [13]. To this end, 156 g of AlCl_3_·6H_2_O (*Fluka*) were dissolved in 137 mL of distilled water and 200 mL ethanol (*Bioenergie Icking GmbH*). After complete dissolution, 10.2 g of oxalic acid (dihydrate, *Merck*) were added and dissolved. Without delay, the reaction mixture was placed in a flask surrounded by an ice bath and cooled down to 4 °C, then 140 mL of propylene oxide (*Acros Organics*) were added, all at once, under vigorous stirring. The ice bath was removed after 6.5 min, and the reaction mixture was stirred at room temperature for another 6.5 min. The obtained sol was poured into a beaker, sealed and placed in an oven at 40 °C for gelation. To ensure a uniform heat transfer, the beaker resided in a water bath. After 24 h, solvent exchange to pure ethanol was carried out under static conditions for 3 days. The opaque white gel was then transferred to a drying cabinet pre-heated at 70 °C, with the lid removed. Drying was terminated after 7 days. Ultimately, the obtained xerogel was pre-calcined at 650 °C for 6 h, with a heating rate of 3 K min^−1^. Characterization of the starting material was carried out as described in Section 3.1.

### 2.2. Mn-Assisted Conversion into α-Al_2_O_3_

The pre-calcined starting material was impregnated with a 1 M aqueous solution of Mn(NO_3_)_2_·4H_2_O (*Alfa Aesar*). Samples were completely immersed in the manganese precursor solution and outgassed at 80 mbar for 1 h. When no more air emerged from the alumina pores, excess solution was removed via filtration and the impregnated samples were then dried in a drying cabinet at 120 °C for 18 h. Finally, Mn(NO_3_)_2_ was thermally converted into manganese oxide at 200 °C for 6 h.

Conversion into α-Al_2_O_3_ of both Mn-impregnated and pristine alumina samples was attempted at different temperatures ranging from 750 to 1200 °C. The heating rate was set to 3 K min^−1^, while dwell times varied from 6 to 48 h. After calcination, excess Mn species from the impregnated samples were removed by stirring in concentrated HCl (37 wt.-%) for 4 h at room temperature.

Impregnated samples are designated “Mn” and reference samples as “Ref”, followed by the calcination conditions. For instance, sample Mn-800-12 is a Mn-impregnated sample calcined at 800 °C for 12 h and leached with HCl. Samples that are investigated before leaching are marked with * (e.g., Mn-800-12*).

To verify thermal stability, a Mn-800-12 sample and a reference sample Ref-650-6 were exposed to a temperature of 750 °C for another 7 days. These are designated as Mn-800-12@750-172 and Ref-650-6@750-172, respectively.

### 2.3. Characterization Techniques

Nitrogen sorption (*Autosorb iQ*, Quantachrome) was used to determine the specific surface area A_BET_ of reference and corundum samples. Prior to analysis, the samples were dried and activated at 250 °C for 10 h under ultra-high vacuum. The determination of specific surface areas was conducted using the linearized form of the BET equation in the range of 0.05 ≤ p p_0_^−1^ ≤ 0.30.

For mercury intrusion, Pascal 140 and Pascal 440 (Porotec) were used. In Pascal 140, the samples were evacuated to 0.2 mbar and then filled with mercury. In Pascal 440, the intrusion measurement up to 400 MPa was performed at room temperature. The contact angle of mercury was set to 140° and the surface tension to 0.48 N m^−1^.

For X-ray powder diffraction analysis (XRD), the samples were finely ground before measuring. The measurement was performed on a STADIP instrument from STOE & Cie GmbH with a Mythen1K detector. Cu-K_α_-radiation was used at 40 kV and 40 mA. Phase identification was carried out in Match! Software (Crystal Impact).

Magic angle spinning (MAS) ^27^Al-NMR spectra were recorded on an Avance-III 400-MHz WB NMR spectrometer (Bruker BioSpin, Rheinstetten, Germany) equipped with a 4 mm MAS BB/^1^H probe at a Larmor frequency of 104.26 MHz. All spectra were collected at a spinning frequency of 10 kHz at 20 °C and referenced externally on the Xsi-scale to water at 4.7 ppm. For magic angle spinning (MAS) experiments, 1024 scans were accumulated with a recycle delay of 0.1 s. All samples were ground and subsequently transferred to the MAS rotor before measuring.

The manganese content of selected synthesized materials was determined using inductively coupled plasma optical emission spectrometry (ICP-OES) with a Perkin ElmerOptimal 8000.

Scanning electron microscopy (SEM) images were obtained using a Leo Gemini 1530 by Zeiss with an Everhart-Thornley detector for collecting secondary electrons. EDX measurements were also carried out on said device, using a SUTW-Sapphire detector. Samples were fixated on a carbon foil and vapor coated with a gold film. The accelerating voltage was 20 kV for both devices.

## 3. Results and Discussion

### 3.1. Properties of the Starting Material

Alumina starting materials for all tests were taken from a single sol-gel batch, which was prepared as described above and previously reported [13]. Reproducibility issues should thus be circumvented. The alumina product was characterized after calcination at 650 °C for 6 h. XRD and ^27^Al-MAS-NMR indicates no long-range order, as is characteristic for transition alumina such as γ-Al_2_O_3_ [10]. A specific surface area A_BET_ of 276 m^2^ g^−1^ was determined by nitrogen sorption. A modal mesopore diameter of 11 nm with a corresponding mesopore volume of 0.55 cm^3^ g^−1^ was established via mercury intrusion. In the SEM images shown in Figure 1, interconnected mesopores embedded in a macropore structure can be identified.

### 3.2. Phase Identification of α-Al_2_O_3_ by XRD and NMR

The pre-calcined alumina material was subjected to calcination at different temperatures with varying dwell times. The temperature range spanned from 750 to 1150 °C, depending on the nature of the samples (pristine or impregnated with manganese precursor solution). Based on the applied calcination temperature, different aluminum phases can be synthesized. While the conversion of γ-Al_2_O_3_ to α-Al_2_O_3_ in general requires temperatures above 1100 °C [24,25,26], the necessary transition temperature is reduced by Mn-impregnation. As a result, α-phase transition can be achieved under much milder conditions [11]. Afterward, Mn-impregnated samples were leached for Mn removal. Conversion of transitional alumina into corundum was monitored by XRD. An error of up to 5 wt.-% for undetectable alumina modifications of lesser crystallinity, e.g., γ, θ, η or κ-alumina is implied. Emerging α-Al_2_O_3_ crystallites only contribute to the diffraction intensity once they surpass the threshold of ≈30 Å, marking the detection limit of XRD. This reservation being stated, complete transition to α-Al_2_O_3_ was observed at 850 °C for a dwell time of 6 h, as the diffraction patterns shown in Figure 2 reveal.

In the reference samples, similar crystallinities can only be obtained by a calcination above 1100 °C at equal calcination times of 6 h (Figure 3). For manganese impregnated samples, conversion attempts at lower temperatures remained unsuccessful up to 750 °C, as the diffraction pattern in Figure 2 demonstrates. Increasing the calcination temperature to 800 °C results in corundum reflexes with very low intensity. However, a substantial growth of both the α-Al_2_O_3_ fraction and its individual crystallites takes place during the following 6 h. Extending the dwell time to 12 h at 800 °C leads to the complete conversion into corundum. To the best of our knowledge, these are the mildest calcination conditions reported to date in obtaining α-Al_2_O_3_ by excluding diaspore-derived corundum [21,22,23]. For comparison, the calcination of non-impregnated alumina at the same conditions (Ref-800-12, Figure 3) leads to a diffraction pattern of an amorphous material, with the major reflexes of γ-Al_2_O_3_ emerging as extremely broad and of very low intensity.

Hence, by the Mn-impregnation, the α-transition threshold temperature can be lowered to 800 °C which is even lower than the 900 °C reported earlier by our group [10]. The reason for this further reduction in the α-transition temperature may be found in a more thoroughly performed impregnation step, and therefore in higher amounts of manganese oxide species on the converted samples.

To further investigate the Mn impregnation, XRD, ICP-OES and SEM-EDX were performed before and after acidic leaching. The different analytical methods were used to determine the successful impregnation and removal of Mn and the subsequent purity of the product. In Figure 3, the XRD pattern of sample Mn-800-12* before leaching depicts well-crystallized bixbyite Mn_2_O_3_ and corundum. The bixbyite pattern vanishes after leaching for 4 h in concentrated HCl (Mn-800-12), implying a complete removal of Mn_2_O_3_. However, ICP-OES analysis (Table 1) reveals that 4.0 wt.-% of the initial 17.6 wt.-% of manganese remain in the leached Mn-800-12. Those findings are reinforced by SEM-EDX whereby 5.5 wt.-% of manganese is found on the materials surface after leaching. The discrepancy in manganese content between the ICP-OES and SEM-EDX is attributable to the different analysis methods. While the former method (ICP-OES) is a bulk analysis, SEM-EDX can only penetrate the surface (≈1 µm) of the sample. As Mn is applied solely by impregnation, it will necessarily be present in higher concentrations near the surface rather than in the bulk (as shown for Mn-800-12*, Table 1).

Although XRD depicts no crystalline manganese phases after leaching (Figure 3), ICP-OES and SEM-EDX show that a Mn content of 3 to 4 wt.-% is always retained. The value cannot be undercut, regardless of the harshness of the leaching conditions, even when boiling aqua regia is used [10]. This observation is very important regarding two aspects, Firstly, it seems to fortify the mechanism postulated by Tsyrulnikov et al. [11], hypothesizing the permanent integration of Mn^3+^-ions into the hexagonal α-Al_2_O_3_ structure. Secondly, the integration of a certain percentage of Mn^3+^-ions into the corundum structure appears to be unavoidable, as previous studies have already suggested [27].

Besides XRD, ^27^Al-MAS-NMR spectroscopy was used to elucidate the structure of the present samples. As depicted in Figure 4, non-impregnated materials synthesized below 1100 °C (Ref-800-12, Ref-1050-6) exhibit two isolated main peaks, with high intensities at 9 and 69 ppm. In addition to the main peaks, symmetric spinning sidebands can be observed at an interval of ~100 ppm (due to the set spinning frequency of 10 kHz). The most intense peak at 9 ppm is assigned to octahedrally coordinated aluminum (θ-Al_2_O_3_), while the less intense peak at 69 ppm can be assigned to either γ- or δ-Al_2_O_3_. Based on the phase transformation of γ- to δ-Al_2_O_3_ around 650 °C, we assume that primarily the δ-phase is present in the samples. Compared to the XRD findings in Figure 3, where XRD amorphous patterns (Ref-800-12) or reflexes with very low intensities (Ref-1050-6) were depicted, ^27^Al-MAS-NMR spectroscopy provided much more detailed information on the present alumina phases, as suggested by O’Dell et al. [18]. At higher temperatures of 1100 °C (Ref-1100-6), a single main peak at 13 ppm and the associated spinning side bands can be observed, which is characteristic for ordered α-Al_2_O_3_ [15]. The sample Mn-800-12 shows a similar periodic pattern, with an even higher intensity. Hence, the synthesis of highly ordered corundum at 800 °C by Mn-impregnation can also be confirmed by ^27^Al-MAS-NMR spectroscopy. In addition, it was shown that no other aluminum species is present, and thus a complete conversion to corundum at 800 °C has taken place.

### 3.3. Porosity and Long-Term Stability of Al_2_O_3_

The pre-calcined γ-alumina reference Ref-650-6 exhibits a specific surface area A_BET_ of 276 m^2^ g^−1^. It decreases to 15 m^2^ g^−1^ after calcination at 1100 °C (Ref-1100-6), and even to 8 m^2^ g^−1^ after calcination at 1150 °C without impregnation. In comparison, a large surface area of 56 m^2^ g^−1^ can be maintained for the Mn-impregnated sample Mn-800-12. This is an unprecedented value for α-Al_2_O_3_, surpassing all previously reported examples [10]. In other publications, sol-gel synthesized α-Al_2_O_3_ has values around 10 to 20 m^2^ g^−1^, much lower than the newly synthesized material [12,28,29]. However, with increasing calcination durations, the specific surface area decreases, as Figure 5 illustrates. During the first 12 h, the mesopore diameter increases from 11 to 20 nm and the mesopore volume decreases from 0.55 cm^3^ g^−1^ to approximately 0.1 cm^3^ g^−1^. Correspondingly, the specific surface area decreases, due to the transformation of γ-Al_2_O_3_ into α-Al_2_O_3_. (cf. Figure 3 and Figure 4 in Section 3.2). With prolonged calcination times (>12 h), the mesopore diameter increases further, with no significant change in the pore volume, resulting in lower specific surface areas. The associated nitrogen sorption isotherms are given in Appendix A.

As suggested by those findings, thermal stability seems to be a significant issue of the synthesized α-Al_2_O_3_. This is already known for diaspore-derived corundum and transition alumina. To further investigate the long-term thermal stability of the synthesized sample Mn-800-12, the material is heated for 7 days at 750 °C. As shown in Table 2 and Appendix A, the thermal treatment resulted in a loss of specific surface area. The A_BET_ decreased by ~48% to 29 m^2^ g^−1^. Interestingly, this decrease in specific surface area was far less pronounced for the starting material that was treated at the same conditions (Ref-650-6@750-172, Appendix A). Here A_BET_ decreased by just ~35% to 180 m^2^ g^−1^. This means that, although porous corundum was synthesized at low temperatures, its porosity is not stable over time at common process temperatures. SEM images given in Figure 6 reveal a densification of the microstructure for Mn-800-12 during the thermal treatment, while primary and secondary particles retain approximately the same size. We assume that this effect contributes to the reduction in specific surface area as well.

## 4. Conclusions

Sol-gel derived γ-Al_2_O_3_ samples were impregnated with a manganese precursor. This led to a considerably lower conversion temperature of 800 °C, compared to 1100–1200 °C, which is usually required to obtain α-Al_2_O_3_. After complete conversion into corundum, a high specific surface area of 56 m^2^ g^−1^ was obtained. Treatment in concentrated HCl led to a removal of Mn, while about 3 wt.-% remained in the sample. Along with the well-established XRD, solid-state ^27^Al-MAS-NMR was introduced as a reliable complementary method to determine the different alumina phases.

We also investigated the long-term thermal stability of the synthesized corundum samples, which is an important issue for any application. After having been exposed to a temperature of 750 °C for 7 days, the specific surface areas decreased by ~48% to 29 m^2^ g^−1^. In conclusion, although porous corundum was synthesized by impregnation with manganese precursors, its porosity does not appear to be stable over time at common process temperatures.

## Figures and Tables

**Figure 1 materials-16-03047-f001:**
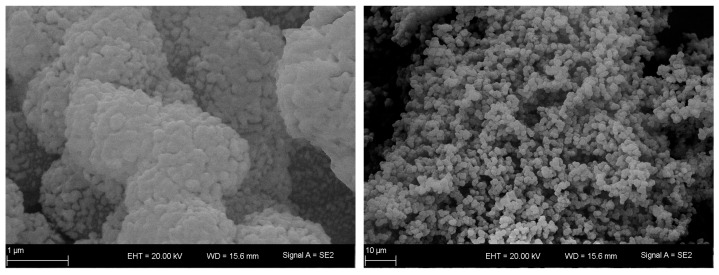
SEM images of the alumina starting material Ref-650-6 depicting its overall morphology (**right**) and in more detail (**left**).

**Figure 2 materials-16-03047-f002:**
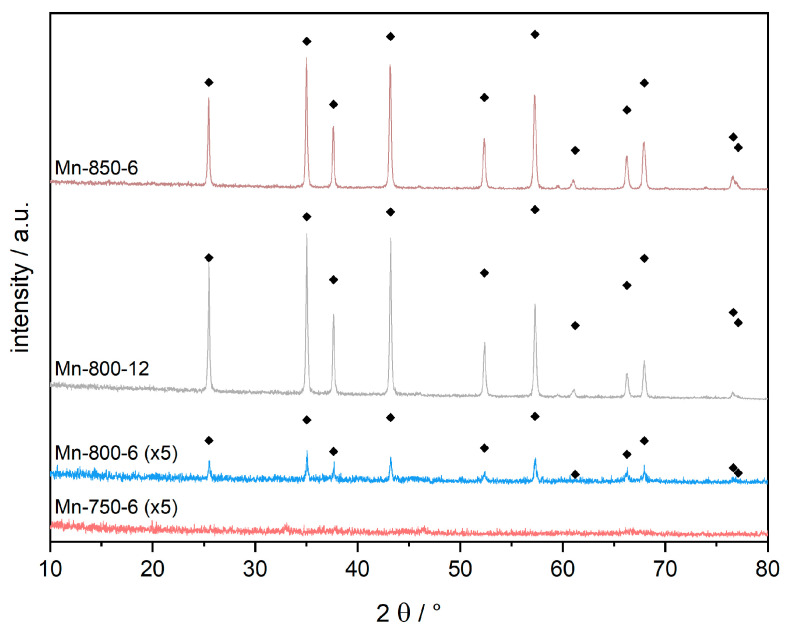
XRD patterns of Mn-impregnated samples after calcination at 750–850 °C for 6 or 12 h. Patterns of samples Mn-750-6 and Mn-800-6 are expanded (×5) for better visibility. Black rhombs mark corundum reflexes.

**Figure 3 materials-16-03047-f003:**
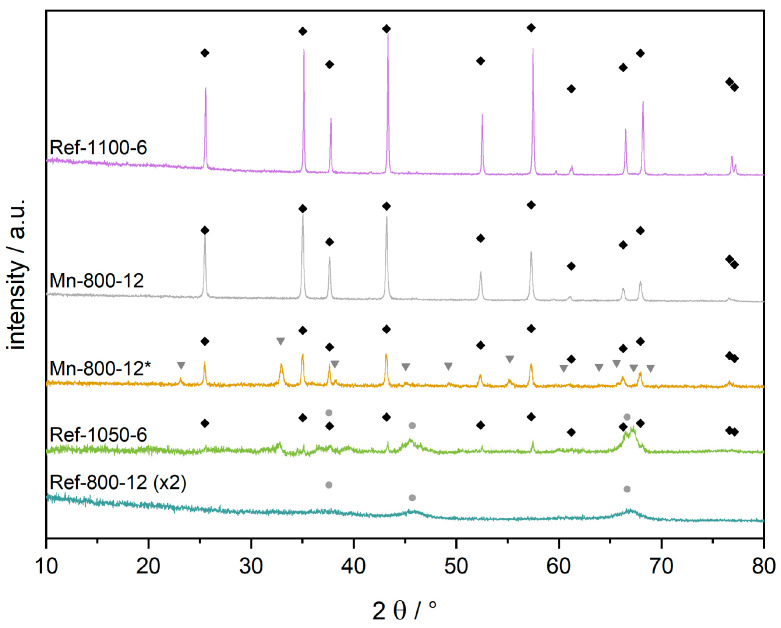
XRD patterns of sample Mn-800-12, calcined at 800 °C for 12 h, before (yellow, marked *) and after (grey) removal of manganese species. For comparison, Mn-free reference samples are given at the top and bottom positions, calcined at 800 °C and 1100 °C, respectively. The pattern of reference sample Ref-800-12 is expanded (×2) for better visibility. Black rhombs mark corundum reflexes, gray triangles show bixbyite (Mn_2_O_3_), and light gray circles indicate the major reflexes of γ-Al_2_O_3_.

**Figure 4 materials-16-03047-f004:**
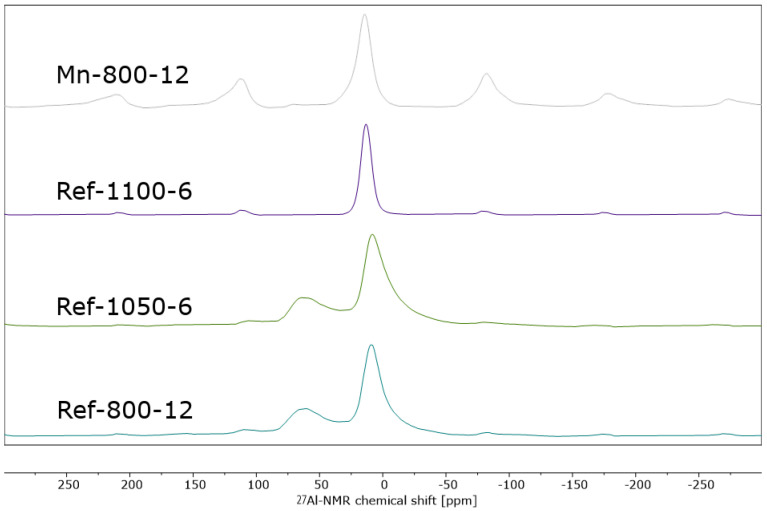
^27^Al-MAS-NMR spectrum of sample Mn-800-12, calcined at 800 °C for 12 h after acid treatment to remove Mn-species. For comparison, spectra of Mn-free reference samples are given, which were calcined at 800 °C for 12 h, 1050 °C for 6 h and 1100 °C for 6 h.

**Figure 5 materials-16-03047-f005:**
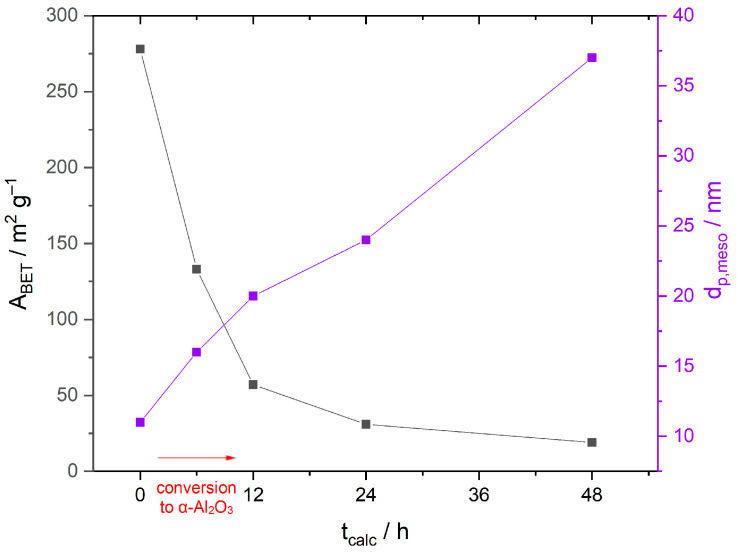
Specific surface areas A_BET_ (grey) and mesopore diameter d_p,meso_ (purple) of Mn-impregnated samples calcined at 800 °C, plotted as a function of the calcination duration. The sharp decrease of A_BET_ within the first few hours of the calcination is due to the transformation of γ-Al_2_O_3_ into α-Al_2_O_3_, as indicated by the red arrow.

**Figure 6 materials-16-03047-f006:**
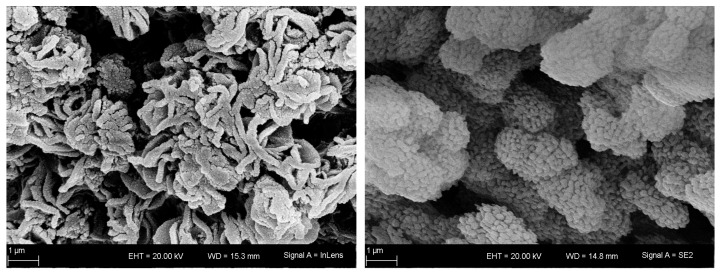
SEM images of samples Mn-800-12 (**left**) and Mn-800-12@750-172 (**right**).

**Table 1 materials-16-03047-t001:** Mn weight-contents of selected samples as determined by ICP-OES and SEM-EDX.

Sample	Mn Weight-Content Determined by…	
	ICP-OES/wt.%	SEM-EDX/wt.%
Mn-800-12	4.0	5.5
Mn-800-12*	17.6	50.7
Mn-900-6	3.0	3.6

**Table 2 materials-16-03047-t002:** Cumulative pore volume, mesopore volume and mesopore diameter obtained by mercury intrusion and specific surface area (A_BET_) obtained by nitrogen sorption.

Sample	A_BET_/m^2^ g^−1^	Pore Diameter/nm	Mesopore Volume/cm^3^ g^−1^	Cumulative Pore Volume/cm^3^ g^−1^
Ref-650-6	276	12	0.51	2.87
Ref-650-6@750-172	180	13	0.51	2.79
Mn-800-6	122	16	0.11	0.61
Mn-800-12	56	20	0.07	0.47
Mn-800-12@750-172	29	31	0.10	0.54
Mn-800-24	31	24	0.10	0.52
Mn-800-48	18	37	0.08	0.72

## Data Availability

The data presented in this study are available on request from the corresponding author.

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
