# Peer review of "Towards High Surface Area α-Al2O3–Mn-Assisted Low Temperature Transformation"

_materials, 2023, doi:10.3390/ma16083047_

Round 1

Reviewer 1 Report

The authors investigated the manganese-assisted synthesis of α-Al2O3 materials with high surface areas from γ-Al2O3 under relatively mild and energy-saving conditions, and characterized their partial physicochemical properties. This work contains some new results and could be considered for publication. However, the authors should revise their manuscript before acceptance for publication according to the following comments:

1.       The authors should determine the residual Mn amount in the as-synthesized α-Al2O3 sample?

2.       How does the Mn assist the conversion of γ-Al2O3 to α-Al2O3? What is the involved mechanism?

3.       It would be better if the authors can provide the figures of N2 adsorption-desorption isotherms and pore size distributions of the samples.

4.   There are some inappropriate English words or expressions in the manuscript. The authors should carefully polish the English of the whole manuscript.

Reviewer 2 Report

Reviewer’s Comments:

The manuscript “Mn-assisted synthesis of α-Al2O3 with high specific surface area” is a very interesting work. In this work, when impregnated with manganiferous precursors, γ-Al2O3 may be converted into α-Al2O3 under relatively mild and energy-saving conditions. In this work, a manganese assisted conversion to corundum at temperatures as low as 800 °C is investigated. To observe the alumina phase transition, in addition to XRD, solid-state 27Al-MAS-NMR is applied. By post-synthetical treatment in concentrated HCl, residual manganese is almost completely removed. Thereby, α-Al2O3 with a very high specific surface area of 56 m2 g1 is obtained after complete conversion.. The results are consistent with the data and figures presented in the manuscript. While I believe this topic is of great interest to our readers, I think it needs major revision before it is ready for publication. So, I recommend this manuscript for publication with major revisions.

1. In this manuscript, the authors did not explain the importance of the surface area in the introduction part. The authors should explain the importance of surface area.

2) Title: The title of the manuscript is not impressive. It should be modified or rewritten it.

3) Correct the following statement “Hence, long-term stability tests were performed at 750 °C for 7 days. These indicated that, although highly porous corundum was synthesized, its porosity decreased over time at common process temperatures”.

4) Keywords: the surface area is missing in the keywords. So, modify the keywords.

5) The authors should explain the following statement with recent references, “The temperature range spanned from 750 to 1150 °C, depending on the nature of the samples (pristine or impregnated with manganese precursor solution).

6) Add space between magnitude and unit. For example, in synthesis “21.96g” should be 21.96 g. Make the corrections throughout the manuscript regarding values and units.

7) The author should provide reason about this statement “To further investigate the influence of the Mn impregnation XRD, ICP-OES and SEM-EDX was performed before and after acidic leaching”.

8. Comparison of the present results with other similar findings in the literature should be discussed in more detail. This is necessary in order to place this work together with other work in the field and to give more credibility to the present results.

9) Conclusion part is very long. Make it brief and improve by adding the results of your studies.

10) There are many grammatic mistakes. Improve the English grammar of the manuscript.

Reviewer 3 Report

This study presents a low temperature transition from γ-Al2O3 to α-Al2O3 as a consequence of Mn-impregnation. The manuscript is well structured and well written. The weak points of this methodology are evidenced in the paper as the impossibility to remove manganese completely from the final material surface and its low thermal stability. This may limit the use of the methodology, especially for the catalytic applications described in the introduction. The combined use of the information from XRD analysis and 27Al-MAS-NMR  for the alumina phase identification is interesting.

Overall, I think this is a solid study and suitable for publication in Materials.

I would ask just a couple of clarifications.

Is it possible to explain why spinning bands are more evident for sample Mn-800-12 with respect to sample Ref-1100-6 in Figure 4? Does it affect the material purity?

a long-term stability investigation was performed also on the reference material? Is the superficial area of the Mn-impregnated still higher than the one of the pristine material after both were exposed to the treatment?
